# Current and future distribution of *Ixodes scapularis* ticks in Québec: Field validation of a predictive model

**Marion Ripoche**[1]*, **Catherine Bouchard**[2], **Alejandra Irace-Cima**[1], **Patrick Leighton**[3], **Karine Thivierge**[4]

**1** Direction des risques biologiques et de la santé au travail, Institut National de Santé Publique du Québec, Montréal, Québec, Canada, **2** Public Health Risk Sciences Division, National Microbiology, Public Health Agency of Canada, Saint-Hyacinthe, Québec, Canada, **3** Faculty of Veterinary Medicine, Department of Pathology and Microbiology, University of Montréal, Saint-Hyacinthe, Québec, Canada, **4** Laboratoire de santé publique du Québec, Institut national de santé publique du Québec, Sainte-Anne-de-Bellevue, Québec, Canada

\* marion.ripoche@inspq.qc.ca

**Data Availability Statement:** The data are owned and available by making a request to Nick Ogden (nicholas.ogden@canada.ca) for predicted data

## Abstract

The incidence of Lyme disease is increasing in Québec and is closely linked to the distribution of *Ixodes scapularis* ticks. A time-to-establishment model developed in 2012 by Leighton and colleagues predicted the year of tick population establishment for each municipality in eastern Canada. To validate if this model correctly predicted tick distribution in Québec, predicted tick establishment was compared to field data from active tick surveillance (2010–2018) using two criteria: i) the detection of at least one tick and ii) the detection of the three questing stages of the tick. The speed of tick establishment and the increase in the exposed human population by 2100 were predicted with the time-to-establishment model. Field observations were consistent with model predictions. Ticks were detected on average 3 years after the predicted year. The probability of tick detection is significantly higher after the predicted year than before (61% vs 27% of collections). The trend was similar for the detection of three tick stages (16% vs 9% of collections). The average speed of tick range expansion was estimated by the model to be 18 km/year in Québec, with 90% of the human population exposed by 2027. The validation of the time-to-establishment model using field data confirmed that it could be used to project *I. scapularis* range expansion in Québec, and consequently the increase in Lyme disease risk over the coming decades. This will help public health authorities anticipate and adapt preventive measures, especially in areas not yet affected by Lyme disease.

## Introduction

Lyme disease, caused by *Borrelia burgdorferi* sensu lato complex (hereafter shortened to *B. burgdorferi*), is the most common vector-borne disease in North America [1]. In Canada, the number of locally acquired human cases per year increased from 144 in 2009 to 2636 cases in 2019 [2]. In Eastern Canada, the incidence of Lyme disease is progressing because of the northward range expansion of the blacklegged tick (*Ixodes scapularis*), the main vector of *B.*

from Leighton et al. (2012) and to Ariane Adam-Poupart (ariane.adam-poupart@inspq.qc.ca) from the Institut national de santé publique du Québec for active tick surveillance data. Interested researchers will need to explain why they need these data according to the process of access of Lyme disease surveillance data in Quebec.

**Funding:** MR was provided with stipend support from by the Public Health Agency of Canada. No funders, except researchers from the Public Health Agency of Canada, had a role in study design, data collection and analysis, decision to publish, or preparation of the manuscript.

**Competing interests:** The authors have declared that no competing interests exist.

*burgdorferi*. This expansion is facilitated by the presence of favorable climatic conditions (temperature and humidity affect tick survival and activity), habitat types, (deciduous and mixed forests are considered suitable habitats for *I. scapularis*) and host communities (white-footed mice–*Peromyscus leucopus* and white-tailed deer–*Odocoileus virginianus* which are the main hosts of *I. scapularis*). Climate change and associated changes in environmental conditions and human activities have contributed to the increasing risk of Lyme disease in Canada [3,4]. In Québec, a province in eastern Canada, the integrated Lyme disease surveillance program provides standardized baseline data on both ticks and human disease to track the spread of Lyme disease [5,6]. Lyme disease has been a notifiable human disease in Québec since 2003. In Québec, the number of locally acquired human cases of Lyme disease has increased from 2 cases in 2008 to 381 by 2019 [7]. Passive tick surveillance programs have existed in Québec since 1990 and consist of submission of ticks found on people or pets, mainly by human and animal health professionals. Active surveillance has been conducted annually in Québec since 2007 and consists of collecting ticks in the environment during spring and summer by the drag sampling method [5,6]. Active surveillance data show that blacklegged ticks were initially only detected south of the St. Lawrence River but are now frequently detected north of the river [8–10].

The distribution of established *I. scapularis* tick populations is a well-known proxy of human Lyme disease risk [11–13]. The tick life cycle typically spans two years with larvae active in summer, nymphs next spring, and adults in autumn [11]. Finding the three stages at one collection site during the same year suggests that they are not of the same generation and provides strong evidence of establishment given that at least one reproduction cycle has occurred locally [12]. Another study suggested that even a single tick collected in the environment during three person-hours of drag sampling provides a strong indicator of local tick establishment [13]. In Québec, public health authorities use the criterion of the three stages in one year to determine if a tick population has established [5]. Currently, ticks can be found throughout Québec due to "adventitious" ticks dispersed by migrating birds or terrestrial animals but established tick populations have only been identified in the southern parts of Québec [14,15]. As Lyme disease risk continues to spread northwards, anticipating and rapidly detecting the location of newly established tick populations would help public health authorities implement local prevention measures.

Active tick surveillance provides useful information about the distribution of ticks, but will not detect and predict all established tick populations in space and time. A predictive risk map of future tick occurrence according to various climate change scenarios was produced based on a temperature threshold suitable for tick establishment [16,17]. Simon *et al.* [18] combined the same model of tick distributions with projected range expansion of white-footed mice to produce a future risk index for *B. burgdorferi* in Québec. These models give interesting information about potential tick distribution in the environment but are too complex to be easily used by public health authorities. Leighton *et al.* [19] took a different approach, analyzing a 20-year time series of passive surveillance data, along with environmental drivers including temperature, elevation, annual rainfall, and local and long-distance dispersal of ticks, to predict the year of future tick establishment for municipalities across eastern Canada. This approach resulted in a predictive map of tick establishment per year with an estimated speed of spread of 46 km/year. This "time-to-establishment" map provides a simple and useful tool for public health authorities to anticipate the change in risk by municipality and by year. Field data from Ontario found good agreement between model predictions and observed presence of *I. scapularis* 10 year later [20], but model projections have yet been evaluated with field data from Québec.

The objective of our study was to assess the concordance between field observations from active tick surveillance (2010–2018) and the timing of tick establishment predicted by the

model of Leighton *et al.* [19]. We investigated how well the model was able to predict 1) the detection of the all three life stages of ticks (strict definition of established tick population), and 2) the detection of at least one tick (less restrictive definition). We then used the model projections to 2100 to investigate the pattern and the speed of tick spread and the potential increase of human exposure to ticks in the coming decades in Québec.

## Materials and methods

### Data

**Study area.** The province of Québec is in the eastern part of Canada, located between the provinces of Ontario to the west and New Brunswick to the east and bordering the United States to the south. It is the largest and the second-most populous province in Canada, with 8,164,361 inhabitants distributed over 1,356,625 km$^2$ [21]. The province is divided into 1,226 municipalities, or census subdivisions (CSDs), with a median area of 100 km$^2$ and an average population of 6,639 inhabitants (ranging from 5 to 1,650,000; median = 1,142). For each municipality included in this study, we estimated the population size and the area (km$^2$) using data from the 2016 census [21].

**Model predictions.** The predicted year of tick establishment for *I. scapularis* for each municipality is the output of a parametric survival regression model developed for eastern Canada passive tick surveillance data (1990–2008) and environmental predictors such as temperature (degree days > 0˚C), elevation, annual rainfall, and local and long-distance dispersal of ticks [19]. For each municipality, there was a mean predicted year, with a lower predicted year (reflecting the fastest progression of ticks) and an upper predicted year (reflecting the slowest progression of ticks), based on the maximum and minimum temperatures observed in each municipality between 1991 and 2008 to account for potential climate change [19].

**Active tick surveillance data.** Tick distribution data come from annual Lyme disease surveillance activities in Québec, carried out jointly by the Institut national de santé publique du Québec (INSPQ), the Ministère de la Santé et des Services Sociaux (MSSS), the Public Health Agency of Canada (PHAC), and the University of Montréal [5,6]. A standardized procedure was used from 2010 to 2018, with drag sampling carried out between June and September in different woodlands that are open to the public. Ticks are generally collected once a year per site but not every year. The sampling period targets nymphs and larvae, but adults can also be collected early in the season. Host-seeking (or questing) ticks were collected by dragging a 1 m$^2$ white flannel cloth over the forest floor, in the morning or early afternoon [22]. Drag sampling did not occur on rainy days or when the forest floor was wet. Dragging method was used following four lines of 500m on either side of a trail, with two lines within 5m of the trail and two lines at 25m of the trail, covering a total of 2000 m$^2$ per site [6]. Every 25 m, the flag was inspected, and collected ticks were sent to the Laboratoire de Santé Publique du Québec (LSPQ) for tick species and life stage identification, using taxonomic keys for larvae, nymphs and adults [23,24].

## Model validation

**Established tick population indicators.** The model developed by Leighton *et al.* [19] was designed to predict the year of tick population establishment, based on a passive surveillance indicator of the detection of at least two adults or one immature tick by active surveillance for two consecutive years. This definition of tick population establishment is not the same as those currently used by Québec's public health authorities. We therefore compared predictions with two active surveillance measures: 1) detection of the three tick stages in the same year, which is the Québec surveillance definition of an established tick population [5] and 2) detection of at

least one tick, which is a broader definition of potential tick population establishment [25]. We examined the lower, mean and upper predicted year of establishment to test the sensitivity of the model to climate variation.

**Statistical analysis.** We investigated the concordance between field observations and model predictions in two steps: firstly, year by year, for each sampling event; secondly, over the study period, aggregating surveillance data from 2020 to 2018 for each municipality

*Sampling event concordance*: First, we investigated the concordance between predicted establishment and tick surveillance data collected during each sampling event (site visit) over the study period, year by year. Do we find tick populations when and where the model predicted? For each sampling event, in a given municipality and year, (n = 444 visits), we determined: i) whether all three tick stages were detected (yes/no), ii) whether at least one stage of tick was detected (yes/no) and iii) whether the model predicted the presence of an established tick population (yes/no) in the municipality on the year of sampling.

*Concordance over the study period*: Secondly, we investigated the concordance between predicted establishment and the presence of ticks in each municipality over the study period, aggregating all tick data from n 2010 to2018. Have we already found tick populations between 2010 and 2018 in municipalities where the model predicted tick population establishment prior to 2018? Because the drag sampling is known to have a low sensitivity to detect tick populations even when these are present [25,26], we aggregated surveillance data by municipality over the study period, cumulating active surveillance data from 2010 to 2018. For each municipality where active surveillance was carried out at least once between 2010–2018 (n = 231 municipalities), we determined: i) whether all three tick stages were detected at least once before 2018 (yes/no), ii) whether at least one stage of tick was detected prior to 2018 (yes/no) and iii) whether the model predicted the presence of an established tick population in 2018 (yes/no).

We compared predicted and observed presence/absence of ticks using the Chi-square test and the kappa coefficient of determination. A significant Chi-square test, with p<0.05, suggests that the distribution of sites with presence of ticks is not at random between areas with predicted or unpredicted established tick population according to the model. We used the matched McNemar test for the analysis of all the sampling because of some repeated samples at the same site. The kappa statistic was interpreted as follows: > 0.75 as excellent, 0.40 to 0.75 as fair to good, and < 0.40 as poor agreement. Analyses were carried out using the *kappa* and *mcnemar.test* functions in R version 4.0.2 [27]. We also calculated the time lag between the exact predicted year and the first detection of all three tick stages or the detection of at least one tick stage in a municipality by active surveillance.

## Projected tick range expansion in Québec: 2019–2100

We used the Leighton *et al.* [19] model to project annual tick range expansion across all Québec municipalities between 2019–2100. The annual speed of tick range expansion in Québec was estimated as the difference between the square root areas of tick establishment in two consecutive years: speed [km/year] = $\sqrt{}$ (area $_{year\ n}$)—$\sqrt{}$ (area $_{year\ n-1}$). From 2006 to 2100, the annual proportion of the Québec population exposed was calculated as the number of people living in municipalities where the model predicted an established tick population divided by the human population of Quebec, based on the 2016 census data [21]. Finally, in order to identify potential regional differences in tick exposure in the coming years, we produced a map of projected tick range expansion to 2100 using the Free and Open Source QGIS software version 3.4.9 (QGIS Geographic Information System).

## Results

### Data

**Model predictions.**   Of Québec's 1,226 municipalities, the model of Leighton *et al.* [19] was able to predict the year of tick population establishment in 1,180 municipalities since 46 (4%) were excluded from the study because of a lack of environmental data. The predicted spread of tick establishment was centrifugal from the southern border of Québec with Ontario and the United States (Fig 1). For 44% of the municipalities, tick population establishment was predicted to occur before 2018 (540 municipalities according to the mean predicted year; 284 and 651 municipalities according to the upper and lower predicted years).

**Active tick surveillance.**   Between 2010 and 2018, ticks were sampled in 231 municipalities, for a total of 444 tick sampling events—municipalities were sampled twice on average during the study period (range: 1 to 7 samples per municipality). At least one *I. scapularis* tick was detected in 218 (49%) site visits, including 52 (12%) with detection of all three stages. In the remaining 226 (51%) site visits, no ticks were detected by the dragging method. At the municipality level, *I. scapularis* was detected in 122 municipalities (53% with at least one tick),

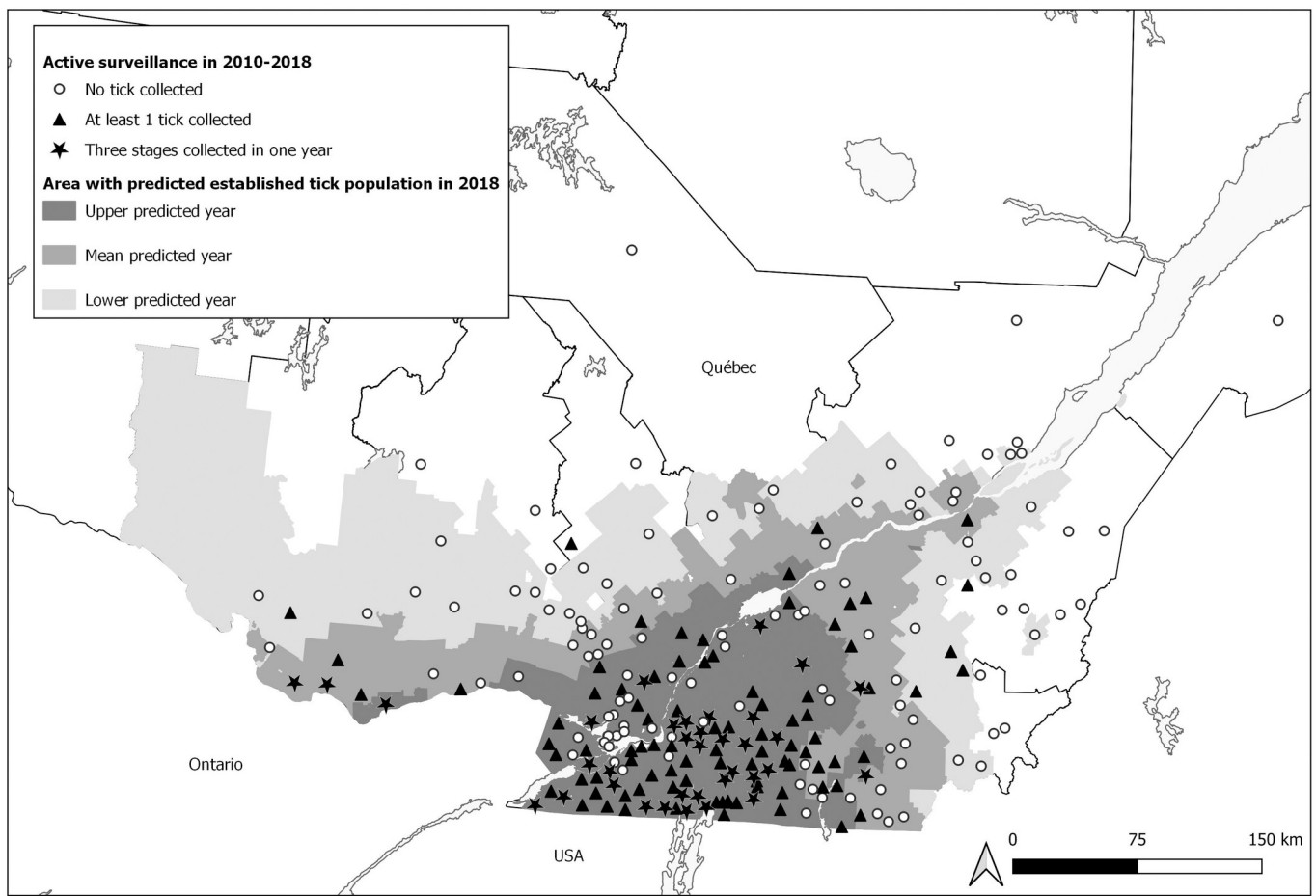

**Fig 1. Tick surveillance in Québec municipalities, 2010–2018, and predicted tick population establishment by 2018.** Active surveillance from 2010 to 2018: municipalities with detection of three stages are indicated by black stars; those with the presence of at least one tick by black triangles and those with no tick detection by white circles. Shaded areas of the map show municipalities with predicted tick population establishment by 2018 according to the lower, mean and upper predicted year (light, medium and dark grey zones, respectively).

including 35 (15%) with detection of the three stages in one year. No ticks were detected in the remaining 109 (47%) municipalities sampled.

## Sampling event concordance

**Detection of three tick life stages.** The probability of detecting the three tick life stages during a single sampling event by active surveillance was significantly higher in municipalities where the model predicted the existence of an established tick population (i.e., sampling conducted in a municipality on or following the predicted year of tick establishment for that municipality). The three stages were detected in 16% (47/289) of the collections that took place after the mean predicted year, but only in 3% (5/155) of the collections that took place before the mean predicted year (lower year: 3% vs 1%; upper year: 16% vs 9%; p<0.001) (Table 1). The upper predicted year provided the best prediction: 64.2% (285/444) of the sampling events were predicted correctly and 35.8% (159/444) of the sampling events were predicted incorrectly (kappa = 0.09 [IC95% 0.01–0.17], p<0.001). The mean predicted year was the second-best prediction: 44.4% (197/444) of sampling events were predicted correctly versus 55.6% (247/444) predicted incorrectly (kappa = 0.09 [IC95% 0.05–0.13], p<0.001). The lower predicted year provided the least accurate prediction: 26.1% (116/444) of sampling events were predicted correctly versus 73.8% (328/444) predicted incorrectly (kappa = 0.03 [IC95% 0.02–0.05], p<0.001) (Table 3). The first detection of the three stages by active surveillance occurred on average three years after the mean predicted year (mean = 3.32; sd = 2.91; range = [-5; 8]) (Table 2). In only one instance were the three stages detected before the lower predicted year, and this was only one year prior to predicted establishment.

**Detection of at least one tick.** Detection of at least one tick showed similar trends as with the detection of all three stages. Ticks were detected in 61% (176/289) of the sampling events that took place on or following the mean predicted year, and 27% (42/155) of tick collections that took place before the mean predicted year (lower year: 52% vs 28%; upper year: 63% vs 41%; p<0.001) (Table 1). The mean predicted year provided the best prediction in this case: 65.1% of sampling events were predicted correctly (289/444) and 34.9% (155/444) of sampling events were predicted incorrectly (kappa = 0.30 [IC95% 0.22–0.38], p<0.001). The upper predicted year was the second-best prediction: 60.8% (270/444) of sampling events were predicted correctly versus 39.2% (174/444) predicted incorrectly (kappa = 0.21 [IC95% 0.12–0.30], p<0.001). The lower predicted year provided the least accurate prediction: 55.4% (246/444) of

**Table 1. Concordance between tick surveillance data and predicted year of establishment.** Observed presence or absence of ticks during active surveillance (no detection, tick presence, three stages) relative to model predictions (before or after the lower, mean and upper predicted year of establishment).

| Predicted year | Established tick population (Year of tick collection) | Number (%) of sampling events (n = 444) | | | | Kappa [IC 95%] A: Tick presence B: Three stages | Chi-square test p-value |
|---|---|---|---|---|---|---|---|
| | | Total | No detection | Tick presence | *Three stages*[1] | | |
| **Lower** | **Expected** (After predicted year[2]) | 378 | 179 (48%) | 199 (52%) | *51 (3%)* | A: 0.11 [0.05–0.18] | <0.001 |
| | **Not expected** (Before predicted year) | 66 | 47 (72%) | 19 (28%) | *1 (1%)* | B: 0.03 [0.02–0.05] | <0.001 |
| **Mean** | **Expected** (After predicted year[2]) | 289 | 113 (39%) | 176 (61%) | *47 (16%)* | A: 0.30 [0.22–0.38] | <0.001 |
| | **Not expected** (Before predicted year) | 155 | 113 (78%) | 42 (27%) | *5 (3%)* | B: 0.09 [0.05–0.13] | <0.001 |
| **Upper** | **Expected** (After predicted year[2]) | 160 | 58 (37%) | 102 (63%) | *26 (16%)* | A: 0.21 [0.12–0.30] | <0.001 |
| | **Not expected** (Before predicted year) | 284 | 168 (59%) | 116 (41%) | *25 (9%)* | B: 0.09 [0.01–0.17] | <0.001 |
| | **Total** | 444 | 226 | 218 | *52* | | |

[1] the number of sampling events with three stages are included in tick presence.

[2] on or following the predicted year.

**Table 2. Time lag between predicted tick establishment and detection of ticks by active surveillance.** Mean, range and standard deviation of the time lag in years between predicted tick establishment and observation of a ticks (three stages or tick presence) in a municipality according to lower, mean and upper predicted year of establishment.

| | Time lag (years) between prediction and observation (mean, range, sd)* | | |
| --- | --- | --- | --- |
| | **Lower year** | **Mean year** | **Upper year** |
| **Three stages** | 5.69 (-1;10) sd = 2.71 | 3.32 (-5;8) sd = 2.91 | -0.19 (-9;5) sd = 3.31 |
| **Tick presence** | 5 (-7;12) Sd = 3.40 | 2.4 (-12;9) Sd = 3.73 | -1.26 (-19;5) Sd = 4.16 |

*Time lag = observation year—predicted year; range = (minimum; maximum); sd = standard deviation.

**Table 3. Concordance between field data and predicted tick establishment by 2018.** Observed presence or absence of ticks during active surveillance (no detection, tick presence, three stages) in municipalities between 2010–2018 vs predicted tick establishment in 2018 (before or after 2018) according to lower, mean and upper model predicted year.

| Predicted year | Established tick pop (Predicted year) | Number (%) of municipalities in 2010–2018 (n = 231) | | | | Kappa [IC 95%] A: Tick presence B: Three stages | Chi-square test p-value |
| --- | --- | --- | --- | --- | --- | --- | --- |
| | | Total | No detection | Tick presence | *Three stages*[1] | | |
| **Lower** | **Expected** (Before 2018) | 209 | 89 (43%) | 120 (57%) | *35 (16%)* | A: 0.17 [0.09–0.25] | <0.001 |
| | **Not expected** (After 2018[2]) | 22 | 20 (91%) | 2 (9%) | *0 (0%)* | B: 0.03 [0.02–0.06] | <0.001 |
| **Mean** | **Expected** (Before 2018) | 171 | 56 (32%) | 115 (68%) | *35 (20%)* | A: 0.43 [0.33–0.54] | <0.001 |
| | **Not expected** (After 2018[2]) | 60 | 53 (88%) | 7 (12%) | *0 (0%)* | B: 0.11 [0.07–0.16] | <0.001 |
| **Upper** | **Expected** (Before 2018) | 118 | 28 (24%) | 90 (76%) | *30 (25%)* | A: 0.47 [0.36–0.59] | <0.001 |
| | **Not expected** (After 2018[2]) | 113 | 81 (72%) | 32 (28%) | *5 (4%)* | B: 0.20 [0.11–0.29] | <0.001 |
| *Total* | | *231* | *109 (47%)* | *122 (53%)* | *35 (14%)* | | |

[1] the number of collections with three stages are included in tick presence.

[2] on or following 2018.

sampling events were predicted correctly versus 44.6% (198/444) predicted incorrectly (kappa = 0.11 [IC95% 0.05–0.18], p<0.001) (Table 3).

The first detection of ticks by active surveillance occurred on average two years after the mean predicted year (presence of ticks: 2.4; sd = 3.73; range = [-12;9]) (Table 2).

## Concordance over the study period

**Detection of three stages.** The probability of detecting the three tick stages by active surveillance between 2010 and 2018 was significantly higher if the model predicted tick establishment before 2018 (Table 3). The three stages were detected in 20% (35/171) of municipalities with a mean predicted year before 2018, but 0% (0/60) of municipalities with a mean predicted year after 2018 (16% vs 0% for lower year and 34% vs 4% for upper year; p<0.001). The upper predicted year was the best prediction: 59.7% (138/231) of the municipalities were predicted correctly and 40.2% (93/231) incorrectly (kappa = 0.21 [IC95% 0.11–0.29], p<0.001). The mean predicted year was the second-best prediction: 41.1% (95/231) of the municipalities were predicted correctly and 58.8% (136/231) incorrectly (kappa = 0.11 [IC95% 0.07–0.16], p<0.001). The lower predicted year was the worst prediction: 24.6% (57/231) of the municipalities were predicted correctly and 75.3% (174/231) incorrectly (kappa = 0.03 [IC95% 0.02–0.25], p<0.001) (Table 3).

**Detection of at least one tick.** Ticks were detected in 68% (115/171) of municipalities with mean predicted year of establishment before 2018 and in 12% (7/60) of municipalities

with mean predicted year after 2018 (74% vs 29% for lower predicted year and 58% vs 10% for upper predicted year; p<0.001). The upper predicted year was the best prediction: 74.0% (171/231) of the municipalities were predicted correctly and 26.0% (60/231) incorrectly (kappa = 0.47 [IC95% 0.36–0.59], p<0.001). The mean predicted year was the second-best prediction: 72.7% (168/231) of the municipalities were predicted correctly and 27.3% (63/231) incorrectly (kappa = 0.43 [IC95% 0.33–0.54], p<0.001). The lower predicted year was the worst prediction: 60.6% (140/231) of the municipalities were predicted correctly and 39.4% (91/231) incorrectly (kappa = 0.17 [IC95% 0.09–0.25], p<0.001) (Table 3).

## Projected tick range expansion in Québec

**Speed of tick spread.** The average speed of projected tick range expansion between 2020–2100 (based on mean predicted year of establishment) was 18 km/year (Fig 2). The speed was similar for upper and lower predicted years (15 and 23 km/year). For the 2020–2030 projections, tick range expansion speed was 22 km/year using the mean predicted year (14 and 21 km/year for the lower and upper years). Speed of range expansion was relatively constant over time, except for two strong accelerations in the 2046–2047 and 2067–2068 projections, which correspond to the prediction of tick population establishment in remote municipalities with large surface areas. Removing these municipalities did not significantly change speed estimate.

**Exposed human population.** According to the model predictions and based on the mean predicted year, 58% of the Québec population were predicted to live in a municipality with an established tick population by 2012, 85% by 2020 and 90% by 2027 (Fig 2). Only one region was concerned in 2008, then 10 regions predicted in 2020 and 16 in 2030 among the 19 regions of Quebec (Fig 3).

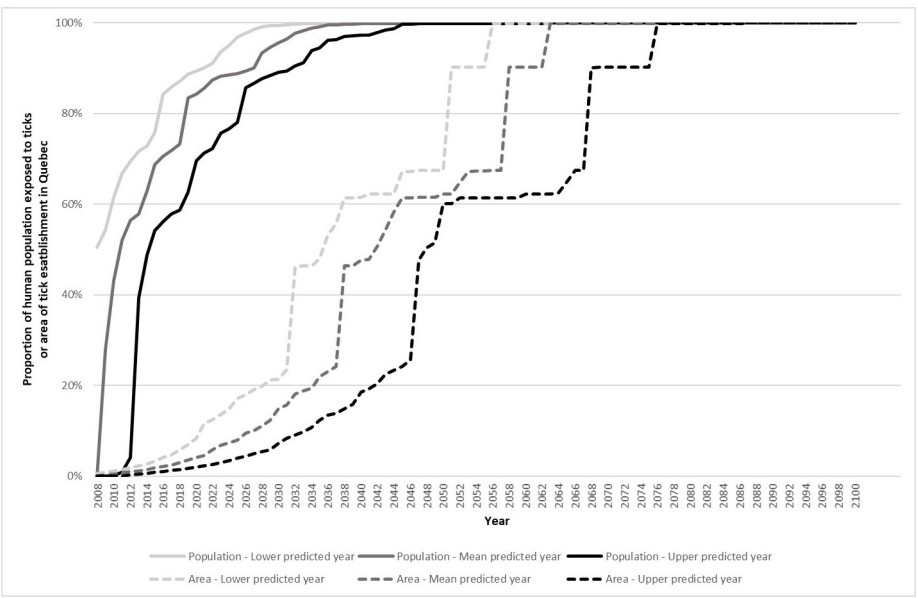

**Fig 2. Predicted increase in the area of tick establishment and the proportion of the human population living in municipalities with established tick population in Québec from 2008 to 2100.** Dashed lines show the percentage of the surface area of the province of Québec predicted to contain an established tick population by the lower, mean and upper predicted years (light, medium and dark grey, respectively). Solid lines show the percentage of the Québec human population living in areas predicted to have an established tick population by the lower, mean and upper predicted years (light, medium and dark grey lines, respectively).

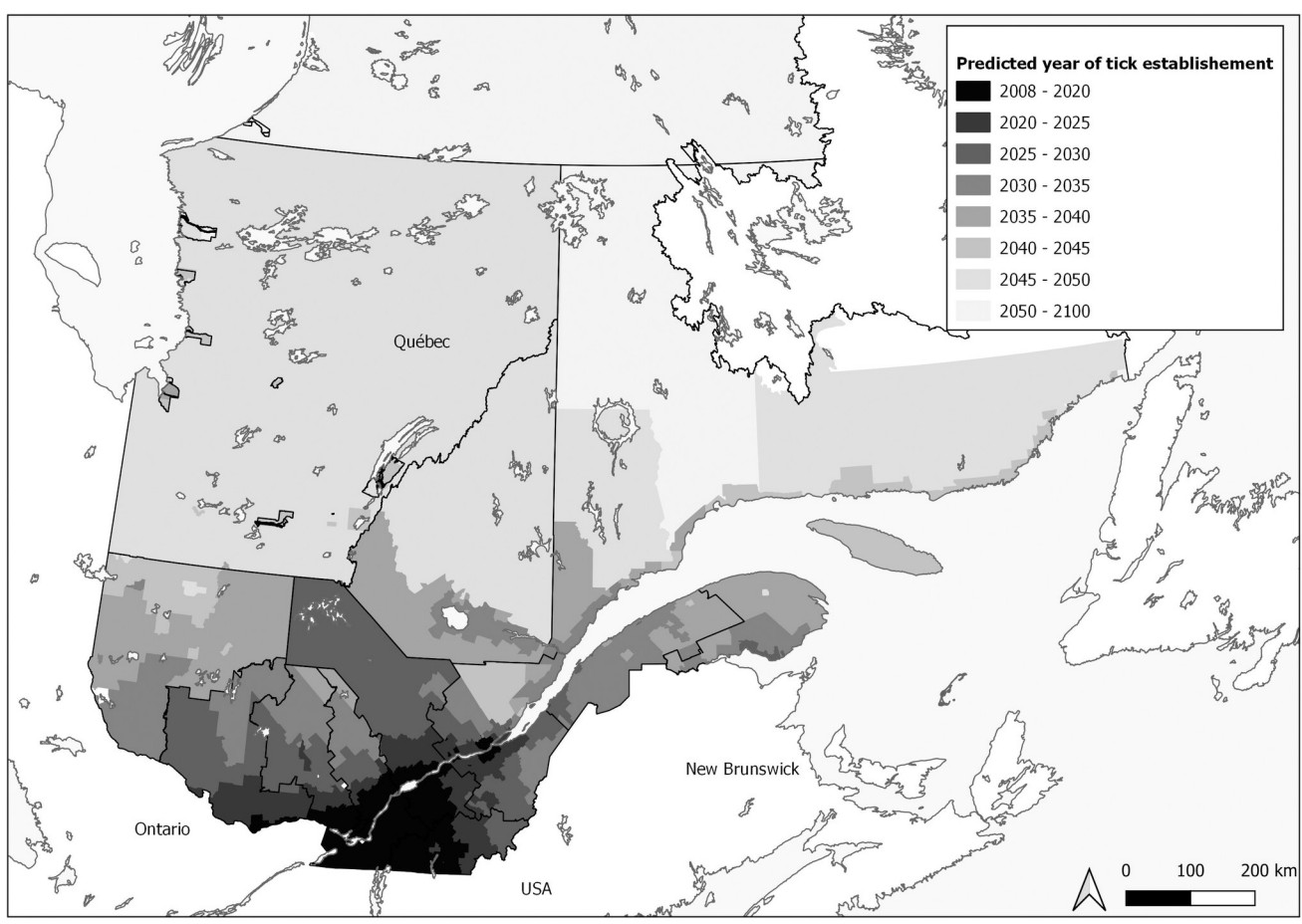

**Fig 3. Prediction of tick population establishment in Québec from 2008 to 2100 by 5-year intervals.** Predicted year of tick population establishment for each Québec municipality based on the upper predicted year of the model of Leighton *et al.* (2012) [19].

**Regional specificity.** Predictions up to 2100 show a northern expansion of tick populations from southern Québec and along the St. Lawrence River (Fig 3). However, in some municipalities, tick establishment varies from surrounding municipalities, with cluster of earlier or later tick establishment. Most regions currently affected by Lyme disease have established tick populations predicted by the model. In the other regions, tick establishment should be expected in the coming years.

## Discussion

In our study, we showed that the "time-to-establishment" model of Leighton *et al.* [19] successfully captured the overall pattern of presence of at least one *I. scapularis* tick and the establishment of a tick population observed through active surveillance in Québec from 2010–2018. Concordance between observations and predictions was relatively low but statistically significant, with no major geographical inconsistencies. The mean predicted year of establishment seems to be a good predictor of tick presence, while the upper predicted year provided a better predictor of the detection of all three tick stages in the field. The predictions correctly reflect the known epidemiological situation in 2018, and the model seems sufficiently reliable to estimate the progression of tick range expansion in the province of Québec, with a predicted

average speed of 18 km/year. The model suggests a continued northward expansion of *I. scapularis*' range in the coming decades, with 90% of Québec's population living in a tick-endemic area by 2027.

### Lyme disease risk is associated with the presence of an established tick population [28–30], but "establishment" remains difficult to define

An important goal for public health authorities is to predict where and when tick populations will establish in order to inform and protect human populations. Established tick populations reproduce locally, but different criteria have been used to operationally define "establishment" based on surveillance data. The first Canadian definition of established tick populations was the detection of all three tick life stages for at least two consecutive years [12]. However, this definition required significant human and financial resources to repeatedly collect ticks in the environment over a two-year period. In Québec, the province has used the detection of all three tick stages within the same year [6] to define an established tick population. However, Québec also used the criterion of "the presence of at least one tick" to determine Lyme disease risk level by municipality [5]. In contrast, the model developed by Leighton *et al.* [19] was designed to predict the year of tick population establishment based on a passive surveillance indicator of the detection of at least two adult ticks or one immature tick by active surveillance for two consecutive years. It was therefore important to compare these two definitions with the model predictions in order to understand how to use this model in public health in Québec.

### Concordance between field data and model predictions is low but statistically significant, probably as a consequence of limits inherent to active surveillance and modelling tools

On the one hand, the sensitivity of drag sampling for detecting ticks in the environment is low because the efficiency of dragging depends on meteorological conditions during sampling [13,26] and because ticks have a heterogeneous distribution at a fine geographical scale [30–34]. Detection of *I. scapularis* larvae, nymph and adults during the same visit is highly unlikely since they are generally active during different seasons [11]. Not observing a tick during a single visit to a municipality may also lead to the incorrect conclusion of absence. On the other hand, detection of any tick stage, especially if only one tick is found, does not always indicate an established tick population. Instead, such observations could be adventitious ticks carried by vertebrate hosts (migratory birds, rodents, deer) into areas without established tick populations [32,35,36]. In our active surveillance data, for 30% of municipalities with a first tick detection, ticks were not systematically detected during sampling in subsequent years. These limits likely explain the low concordance we observed: with an agreement of 65% between predictions and observations, the model is better than flipping a coin to predict tick presence or detection of all three stages in active surveillance in a given municipality and year, but shows better agreement (74%) when cumulating surveillance data over the study period. In fact, multiple sampling events for each municipality reduce the sampling error due to the failure of detecting ticks at a site where they are in fact established. These limits also explain the better results with the criterion of "presence of at least one tick" rather than the criterion of "presence of three stages", which is harder to attain with active surveillance methods in Québec. Moreover, because of limitations in logistical capacity, active surveillance is not carried out every year in all municipalities in Québec; this incomplete sampling underestimates tick establishment and contributes to the time lag between the predicted year of establishment and detection of ticks by field sampling (three years on average in our study).

## Model predictions provide an accurate overall portrait of tick range expansion, and fine-scale variation in local establishment is expected in areas of emerging risk

Some divergence between predictions and field data may be explained by factors that influence local tick establishment such as forest type, micro-climate, and distribution and behavior of vertebrate hosts. These factors were not included in the model of Leighton *et al.* [19]. This could explain some divergence between predictions and observations, as will also influence the real progression of ticks over the coming decade. Temperature is thought to be the most important factor determining suitable areas for tick establishment [16,37], but other factors could play an important in northern Québec where the habitat is dominated by coniferous forest, considered to be marginal habitat for ticks, and where vertebrate host communities are different from southern Québec. Leighton *et al.* [19] suggested that rainfall and elevation may also influence the speed of tick population establishment. Moreover, previous studies conducted at finer spatial scales highlighted the heterogeneous distribution of ticks [30–34], probably as a result of differences in local environmental conditions and host communities that support local and long-distance dispersal of ticks. At the Québec scale, the predicted established range of the tick in 2018 correctly overlaps the confirmed distribution of ticks based on active surveillance (presence or three stage), suggesting that model predictions are reliable at this scale. In addition, the small number of municipalities with detection of the three tick stages that are located outside of the predicted area of establishment are situated very close to the predicted area. Moreover, we noticed local particularities in the predictive map (e.g., zones with tick establishment earlier or later than surrounding area) which were also observed in the acarological surveillance data (results not presented here). Overall, the model successfully predicted the progression of tick range expansion over the past decade, and observed fine-scale variation in local establishment expected within this emerging risk area. Further studies could improve the existing model and refine predictions by integrating more recent environmental and surveillance data.

## Because field observations were consistent with model predictions, we explored the speed at which tick population establishment progressed across Québec

According to the model, the overall speed of expansion of the established range of *I. scapularis* in Québec was estimated to be 18 km/year [95% CI; 15–23 km/year], which is lower than the value of 46 km/year for eastern Canada suggested by Leighton *et al.* [19]. The original model was for all eastern Canadian provinces (i.e., Québec, Ontario, etc.), but differences in the speed of range expansion among provinces were not explored in that study. A recent study carried out in southern Ontario corroborated the speed of 46 km/year [20], by comparing the predicted year of establishment of a tick population within a radius of 46 km from a site with an established tick population; however, the authors did not calculate the speed predicted by the model in Ontario as was done here. Using a model based on degree days $> 0°C$ [16,17] and tick abundance, Simon *et al.* [18] estimated a northward expansion of *I. scapularis* of 300 km from 2011 to 2050 in southern Québec, representing a speed of tick range expansion of 7 km/year, which is more consistent with the estimate from the present study. The model by Simon *et al.* [18] was based on environmental variables and was not biased by administrative boundaries. In our study, because we calculated speed based on the difference between the total areas of municipalities with established tick populations area in two consecutive years, larger municipalities, especially in northern Québec, can artificially increase estimates of the speed of tick progression. The real speed of establishment also depends on suitable habitat and the

composition of host communities, and won't be consistent over time and space even within a municipality. Finally, beyond the general acceleration of range expansion expected with warming climate conditions [19], climate change is likely to impact the observed speed of tick range expansion in the coming decades through its multiple effects on the different components of tick ecology.

### Application in public health

Despite the aforementioned limitations, the Leighton *et al.* [19] model reliably estimated the recent pattern of tick range expansion in Québec, and by extension the risk of human Lyme disease, providing a useful projection of risk for the coming decades. The presence of at least one tick detected by active surveillance is a criterion used to identify "at-risk" municipalities in Québec [5] and may reflect the presence of newly established tick populations. The detection of the three tick stages during the same season by active surveillance, with detection of *B. burgdorferi* in at least one nymph, reflects the presence of an established tick population and is used to identify "endemic" municipalities in Québec [5]. Consequently, the mean predicted year of establishment could be a useful indicator "at-risk areas" while the upper predicted year could be used as an as indicator of "endemic areas", currently and for the coming decades. This approach helps to estimate the present and future risk levels of all municipalities in Québec, and could guide future active surveillance activities. Since the distribution of *B. burgdorferi* is expected to follow a similar trajectory to that of *I. scapularis* [22], the projection maps could help anticipate future changes in the epidemiology of Lyme disease in Québec. The distribution of tick also provides an estimate of the exposed human population, with potentially 90% of Québec's population living in an endemic area by 2027, which could be refined using the demographic projections for Québec. Even though other factors influence the number of human cases of Lyme disease (e.g., human behavior, prevention, diagnosis) [4], reliable projections of future risk could nevertheless help public health authorities develop preventive measures, particularly in regions not yet affected by Lyme disease.

### Conclusion

Our study demonstrated that a predictive model of tick range expansion can be used to provide reliable projections of changes in the distribution of tick-endemic areas over a 10-year period, and useful estimates of the changing risk of Lyme disease at the municipality level in Québec for the coming years. Assessing the concordance between a predictive model and field data is an important step in evaluating such prediction tools for use by public health authorities. This model should be updated as new field surveillance data become available in Québec in the coming years, and could be similarly validated for use in predicting future spread of ticks and Lyme disease risk within other Canadian provinces.

### Supporting information

**S1 File.**
(DOC)

### Acknowledgments

We thank all the people involved in tick collection in the field as part of the active tick surveillance program. We would also like to thank Julie Ducrocq and two anonymous reviewers for their helpful feedback on the manuscript.

## Author Contributions

**Conceptualization:** Marion Ripoche, Catherine Bouchard, Alejandra Irace-Cima, Patrick Leighton.

**Data curation:** Marion Ripoche, Patrick Leighton, Karine Thivierge.

**Formal analysis:** Marion Ripoche.

**Funding acquisition:** Alejandra Irace-Cima.

**Investigation:** Marion Ripoche.

**Methodology:** Marion Ripoche, Catherine Bouchard, Patrick Leighton.

**Writing – original draft:** Marion Ripoche.

**Writing – review & editing:** Marion Ripoche, Catherine Bouchard, Alejandra Irace-Cima, Patrick Leighton, Karine Thivierge.

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
