## [Decision Letter · Decision Letter 0]

3 Jun 2021

PONE-D-21-14260

Current and future distribution of Ixodes scapularis ticks in Québec: field validation of a predictive model previously developed in Canada

PLOS ONE

Dear Dr. Ripoche,

Thank you for submitting your manuscript to PLOS ONE. After careful consideration, we feel that it has merit but does not fully meet PLOS ONE’s publication criteria as it currently stands. Therefore, we invite you to submit a revised version of the manuscript that addresses the points raised during the review process.

Both reviewers felt this study is a worthwhile contribution to the field. Please see attached reviewer's comments and pay careful consideration to grammatical errors.

We look forward to receiving your revised manuscript.

Kind regards,

Catherine A. Brissette, Ph.D.

Academic Editor

PLOS ONE

Journal Requirements:

3. We note that Figures 1 and 3 in your submission contain map images which may be copyrighted. All PLOS content is published under the Creative Commons Attribution License (CC BY 4.0), which means that the manuscript, images, and Supporting Information files will be freely available online, and any third party is permitted to access, download, copy, distribute, and use these materials in any way, even commercially, with proper attribution. For these reasons, we cannot publish previously copyrighted maps or satellite images created using proprietary data, such as Google software (Google Maps, Street View, and Earth). For more information, see our copyright guidelines: http://journals.plos.org/plosone/s/licenses-and-copyright.

3.1.    You may seek permission from the original copyright holder of Figures 1 and 3 to publish the content specifically under the CC BY 4.0 license. 

3.2.    If you are unable to obtain permission from the original copyright holder to publish these figures under the CC BY 4.0 license or if the copyright holder’s requirements are incompatible with the CC BY 4.0 license, please either i) remove the figure or ii) supply a replacement figure that complies with the CC BY 4.0 license. Please check copyright information on all replacement figures and update the figure caption with source information. If applicable, please specify in the figure caption text when a figure is similar but not identical to the original image and is therefore for illustrative purposes only.

Reviewers' comments:

Reviewer's Responses to Questions

**Comments to the Author**

1. Is the manuscript technically sound, and do the data support the conclusions?

Reviewer #1: Yes

Reviewer #2: Yes

2. Has the statistical analysis been performed appropriately and rigorously? 

Reviewer #1: Yes

Reviewer #2: Yes

3. Have the authors made all data underlying the findings in their manuscript fully available?

Reviewer #1: Yes

Reviewer #2: Yes

4. Is the manuscript presented in an intelligible fashion and written in standard English?

Reviewer #1: No

Reviewer #2: Yes

5. Review Comments to the Author

Reviewer #1: Title: Current and future distribution of Ixodes scapularis ticks in Québec: field validation of a predictive model previously developed in Canada

Authors: Marion Ripoche, Catherine Bouchard, Alejandra Irace-Cima, Patrick Leighton, Karine Thivierge

Summary of the study: The spread of Ixodes scapularis ticks into Canada and the resultant increase in the incidence of Lyme borreliosis in the Canadian population is a well-documented example of how climate change can increase the public health burden of vector-borne diseases. A model by Leighton and colleagues predicted the speed at which blacklegged ticks would expand their range in eastern Canada and establish new tick populations (1). The purpose of the present study is to test how this model performs in southern Quebec where the authors have spent 9 years (2010 – 2018) doing active surveillance for blacklegged ticks by using the dragging method. The authors conducted 444 tick sampling sessions in 231 census subdivisions (CSDs) between 2010 and 2018. The authors used two different criteria to determine whether a particular site had an established tick population: (1) sampling of all three tick stages (strict definition) and (2) sampling of a single blacklegged tick (less restrictive definition). The authors used the model by Leighton and colleagues to predict the year of blacklegged tick population establishment for each of the sampling sessions and CSDs (they used a mean, lower, and upper estimate). They then compared these predicted values (3 predictions) to the observed values from their active surveillance (2 criteria for an established tick population). The general result was that active surveillance was more likely to find ticks in sites that were predicted to have established tick populations according to the model compared to sites that were not predicted to have established tick populations. For the 444 sampling occasions, the model based on the mean year predicted 65.1% correctly (289/444) and 34.9% (155/444) incorrectly (Table 1). For the 231 CSDs, the model based on the upper year predicted 74.0% correctly (171/231) and 26.0% (60/231) incorrectly (Table 3). The model predicts that 90% of the human population in Quebec will be leaving in an area with an established tick population by 2027 (prediction based on the mean).

Major comments:

-The authors have used a straightforward approach to answering their question, and I don’t have too many criticisms of the study.

-The authors show that active surveillance is more significantly more likely to find ticks in areas that were predicted to have established tick populations compared to sites that were not. However, it would be useful if the authors came up with an overall measure of the ability of their model to predict the correct answer. For example, in Table 1 for the lower predicted year, the model predicted 55.4% of the sampling occasions correctly [(199 + 47)/444 = 246/444] and 44.6% of the sampling occasions incorrectly [(179 + 19)/444 = 198/444]. In other words, for this particular combination of prediction (lower year) and outcome (1 tick present is an established tick population), the model is not much better than flipping a coin. It would be good if this was pointed out in the manuscript. For the 231 CSDs, the model does better and for the upper year, it predicts 74.0% (171/231) of the CSDs correctly and 26.0% of the CSDs incorrectly (60/231). Again, it would be good to point out that for this combination of prediction (upper year) and outcome (1 tick present is an established tick population), the model performs better than a random coin toss.

-In the Discussion, the authors point out that the original model by Leighton and colleagues (1) predicted a speed of progression of established tick populations of 46 km per year. The estimated speed in the current study is considerably slower, 18 km per year over the time period of 2020 – 2100 and 22 km per year over the time period 2020 – 2030. A study in southern Ontario confirmed the speed of progression of established tick populations of 46 km per year, whereas a study in southern Quebec found a speed of progression of 7 km per year. In the Discussion, it is not really clear why the speed of progression differs between Quebec and Ontario, and some more clarification would be helpful.

-In the Discussion, the authors discuss the application of their model for public health. However, according to their model predictions, 90% of the human population of Quebec will be living in an area with an established tick population by 2027. If this prediction comes true, what is there left to say for public health authorities other than that almost the entire population in Quebec is at risk for Lyme disease and ask everyone to be careful?

-The manuscript was formatted with bullets. Is this a new formatting requirement for PLOS ONE manuscripts? If not, please don’t use bullets, as it gives reviewers the impression that the manuscript was hastily prepared.

-The manuscript contained many small grammatical errors. The authors should do a better job of checking the manuscript for such errors before submitting it to a journal. I have attached a Word document with ‘Track changes’ that documents a number of these grammatical errors.

Advice for future studies:

-The authors point out that the dragging method does not always detect the ticks at a particular site, even if the site is believed to have an established tick population. In future studies, the authors should consider using site occupancy models to address this problem. For example, you may sample the same site in 2016, 2017, and 2018, and find ticks in 2016 and 2018, but not in 2017. If we assume that the ticks were there in 2017, but were not detected, we can estimate the probability of not detecting ticks at a site with an established tick population. By separately estimating the probability of detection versus the probability of an established tick population, you will get better estimates of the latter. In Table 1, the best model (based on the mean year) predicts 65.1% of the sampling occasions correctly, whereas in Table 3, the best model (based on the upper year) predicts 74.0% of the CSDs correctly. This comparison suggests that failing to detect ticks in a site where they are established is less of a problem for the CSDs than the sampling occasions. This observation makes sense, because there are multiple sampling occasions for each CSD, which reduces the error due to the failure of detecting ticks at a site where they are established. If you conducted a very large number of sampling occasions at each site (e.g., hundreds), you would eventually have perfect knowledge of which sites had established tick populations and which did not.

-The authors do not give much insight as to why their model is not working better. They point out that a number of factors are not included in the original model by Leighton and colleagues (1). For example, the model is heavily reliant on temperature, but does not include elevation and rainfall. This study would have been more interesting if the authors had dropped components from the original model to quantify the contribution of those components to the accuracy of the predictions. Thus, an unsatisfying feature of this study is that the authors and the readers don’t really learn anything new about how the model by Leighton et al. could be improved. If the authors plan on future studies investigating the predictive ability of the model by Leighton et al., they should consider comparing different sub-models that drop different components of the original model, so that we learn which components of the models are actually important for making the correct predictions.

References

1. Leighton PA, Koffi JK, Pelcat Y, Lindsay LR, Ogden NH. 2012. Predicting the speed of tick invasion: an empirical model of range expansion for the Lyme disease vector Ixodes scapularis in Canada. J Appl Ecol 49:457-464.

Reviewer #2: GENERAL REMARKS

The study validates a predictive model on tick distribution in Canada, by comparing collected field data with the model predictions. These types of model evaluations are useful to assess and improve on the predictive model. However, although I would consider this study to be worthwhile, there are many linguistic errors to be addressed before the publication of this manuscript. Please find more specific remarks below.

ABSTRACT

If the use of title words such as Background, Objective etc. was a conscious decision, these should be clearer. Perhaps adding “:” or a line break can make this distinction between different subheadings more readable.

Line 26: “The incidence of Lyme disease”, or “The recorded number of Lyme disease cases” is increasing in Québec…

Line 27: remove “,” after “2012”

Line 28: put “until 2100” after “establishment”

Line 44: I would add “Ixodes scapularis” to the list of key words

INTRODUCTION

Line 49: Locally acquired cases (per year? In total?) please specify. You could say “increased from 144 annual cases in 2009, to 2636 in 2019”. Put “in Canada” at the beginning of the sentence.

Line 51: “is progressing”

Line 58: check commas and punctuation in general

Line 77: “Active acarological surveillance”

Line 95: all three stages

METHODS

Study area: As this is a study assessing the viability of a model based on climatic data: Some general climatic information (mean yearly temperatures, precipitation), would be welcome. This is also true for the elevational range (as this is factored into the model), and overall vegetation composition (what type of forests).

Some general information on I. scapularis would be welcome, including the vectorial capacity of this species, as well as its ecology (questing behavior and host-parasite relations).

Line 104: “province”

Line 117: tick distribution “data”, comes from …

Line 123: which configuration? Lines, quadrats, one single 2000m line? Specify

Line 130: spacing after citation

Line 131: year of establishment

Line 132: years

Line 139: “tick populations were”, or “a tick population was” supposed to be established

Line 138: remove first “of the model”, remove first comma

Line 146: “A tick population” or “tick populations”, there where and when…

Line 147: Predicted for it

Line 151: “if a tick population is predicted to be established in the area, then ticks are also expected to be found in the field data”

Line 169: for clarity, I suggest inserting a full equation in word in the form of:

Speed[km/year]=√(area*year*n)- √(area*year*n-1)

Line 172: “exposed to ticks”, remove first comma,

Line 173: “predicted an established tick population”

Line 175: “across the population of Quebec”

Line 176: “…of established tick populations”, “was produced/generated”

RESULTS

Line 184: add percentage

Line 199: “twice on average”

Line 209: “Concordance per collection”

Table 1: Is there a reason why the three stages column data is in italic? If so, specify.

Table 2: If you use “Sd” in the table, the *text should be “Sd” as well, not “sd”.

Table 1 and 3: footnote: “are included”

Line 240: remove second “detection”

Line 242: “the three stages”

Line 245-246: If you include the statistical result, please also include the p-value

DISCUSSION

I am missing some recommendations on if/how the findings of this study (field validation) could be used to enhance the model developed by Leighton (2012).

Line 298: “time-to-establishment”

Line 310: where and when “A tick population” or “tick populations” will establish

Line 315: Collect ticks

Line 320: Years

Line 321: remove comma. Risk of what? Specify Lyme disease risk

Line 322: “with the model predictions”

Line 338-339: Because of “limitations” in “logistical” capacity

Line 349: “influential factor”

Line 350: northern, sometimes “Quebec” is written “Québec”. Please be consistent.

Line 351: host communities are different how?

Line 352: “A previous study”, or “previous studies”

Line 363: remove comma

6. PLOS authors have the option to publish the peer review history of their article (what does this mean?). If published, this will include your full peer review and any attached files.

Reviewer #1: No

Reviewer #2: No

---

## [Author Response · Author response to Decision Letter 0]

16 Oct 2021

Dear reviewers,

Thanks for your pertinent comments and suggestion, it was a real contribution for the manuscript. Thank you for your understanding about my English, which is unfortunately far from perfect, and for making the effort to overpass or correct it by reviewing the article, it is much appreciated.

You will find below and in the revised manuscript the response to your comments.

Review Comments to the Author

Reviewer #1: Title: Current and future distribution of Ixodes scapularis ticks in Québec: field validation of a predictive model previously developed in Canada

Authors: Marion Ripoche, Catherine Bouchard, Alejandra Irace-Cima, Patrick Leighton, Karine Thivierge

Summary of the study: The spread of Ixodes scapularis ticks into Canada and the resultant increase in the incidence of Lyme borreliosis in the Canadian population is a well-documented example of how climate change can increase the public health burden of vector-borne diseases. A model by Leighton and colleagues predicted the speed at which blacklegged ticks would expand their range in eastern Canada and establish new tick populations (1). The purpose of the present study is to test how this model performs in southern Quebec where the authors have spent 9 years (2010 – 2018) doing active surveillance for blacklegged ticks by using the dragging method. The authors conducted 444 tick sampling sessions in 231 census subdivisions (CSDs) between 2010 and 2018. The authors used two different criteria to determine whether a particular site had an established tick population: (1) sampling of all three tick stages (strict definition) and (2) sampling of a single blacklegged tick (less restrictive definition). The authors used the model by Leighton and colleagues to predict the year of blacklegged tick population establishment for each of the sampling sessions and CSDs (they used a mean, lower, and upper estimate). They then compared these predicted values (3 predictions) to the observed values from their active surveillance (2 criteria for an established tick population). The general result was that active surveillance was more likely to find ticks in sites that were predicted to have established tick populations according to the model compared to sites that were not predicted to have established tick populations. For the 444 sampling occasions, the model based on the mean year predicted 65.1% correctly (289/444) and 34.9% (155/444) incorrectly (Table 1). For the 231 CSDs, the model based on the upper year predicted 74.0% correctly (171/231) and 26.0% (60/231) incorrectly (Table 3). The model predicts that 90% of the human population in Quebec will be leaving in an area with an established tick population by 2027 (prediction based on the mean).

Major comments:

-The authors have used a straightforward approach to answering their question, and I don’t have too many criticisms of the study.

MR:ok

-The authors show that active surveillance is more significantly more likely to find ticks in areas that were predicted to have established tick populations compared to sites that were not. However, it would be useful if the authors came up with an overall measure of the ability of their model to predict the correct answer. For example, in Table 1 for the lower predicted year, the model predicted 55.4% of the sampling occasions correctly [(199 + 47)/444 = 246/444] and 44.6% of the sampling occasions incorrectly [(179 + 19)/444 = 198/444]. In other words, for this particular combination of prediction (lower year) and outcome (1 tick present is an established tick population), the model is not much better than flipping a coin. It would be good if this was pointed out in the manuscript. For the 231 CSDs, the model does better and for the upper year, it predicts 74.0% (171/231) of the CSDs correctly and 26.0% of the CSDs incorrectly (60/231). Again, it would be good to point out that for this combination of prediction (upper year) and outcome (1 tick present is an established tick population), the model performs better than a random coin toss.

MR:I agree, it is clearer like that, I added that in the result and discussion.

-In the Discussion, the authors point out that the original model by Leighton and colleagues (1) predicted a speed of progression of established tick populations of 46 km per year. The estimated speed in the current study is considerably slower, 18 km per year over the time period of 2020 – 2100 and 22 km per year over the time period 2020 – 2030. A study in southern Ontario confirmed the speed of progression of established tick populations of 46 km per year, whereas a study in southern Quebec found a speed of progression of 7 km per year. In the Discussion, it is not really clear why the speed of progression differs between Quebec and Ontario, and some more clarification would be helpful.

MR: We calculated the speed with the data from the model. I don’t know the predicted speed for Ontario.

-In the Discussion, the authors discuss the application of their model for public health. However, according to their model predictions, 90% of the human population of Quebec will be living in an area with an established tick population by 2027. If this prediction comes true, what is there left to say for public health authorities other than that almost the entire population in Quebec is at risk for Lyme disease and ask everyone to be careful?

MR:It is true. But it is important to explain to region with no established tick population for the moment, that it will be coming in the next decades, even for region in the north with a rude winter

-The manuscript was formatted with bullets. Is this a new formatting requirement for PLOS ONE manuscripts? If not, please don’t use bullets, as it gives reviewers the impression that the manuscript was hastily prepared.

MR:the bullet was to mark the section and sub-section. I deleted them.

-The manuscript contained many small grammatical errors. The authors should do a better job of checking the manuscript for such errors before submitting it to a journal. I have attached a Word document with ‘Track changes’ that documents a number of these grammatical errors.

MR: Thanks for having corrected grammatical errors. I really appreciate that, and I will find a way to check before submission the next time

Advice for future studies:

-The authors point out that the dragging method does not always detect the ticks at a particular site, even if the site is believed to have an established tick population. In future studies, the authors should consider using site occupancy models to address this problem. For example, you may sample the same site in 2016, 2017, and 2018, and find ticks in 2016 and 2018, but not in 2017. If we assume that the ticks were there in 2017, but were not detected, we can estimate the probability of not detecting ticks at a site with an established tick population. By separately estimating the probability of detection versus the probability of an established tick population, you will get better estimates of the latter. In Table 1, the best model (based on the mean year) predicts 65.1% of the sampling occasions correctly, whereas in Table 3, the best model (based on the upper year) predicts 74.0% of the CSDs correctly. This comparison suggests that failing to detect ticks in a site where they are established is less of a problem for the CSDs than the sampling occasions. This observation makes sense, because there are multiple sampling occasions for each CSD, which reduces the error due to the failure of detecting ticks at a site where they are established. If you conducted a very large number of sampling occasions at each site (e.g., hundreds), you would eventually have perfect knowledge of which sites had established tick populations and which did not.

MR: I agree with you, but active surveillance in Quebec is currently limited to one visit per municipality and per year, and aa we are note able to visit all municipality each year, there is a turn over. But we have some sentinels sites, sampled each year, that could be used to estimate the probability of detection or not detection in a site with established tick population.

-The authors do not give much insight as to why their model is not working better. They point out that a number of factors are not included in the original model by Leighton and colleagues (1). For example, the model is heavily reliant on temperature, but does not include elevation and rainfall. This study would have been more interesting if the authors had dropped components from the original model to quantify the contribution of those components to the accuracy of the predictions. Thus, an unsatisfying feature of this study is that the authors and the readers don’t really learn anything new about how the model by Leighton et al. could be improved. If the authors plan on future studies investigating the predictive ability of the model by Leighton et al., they should consider comparing different sub-models that drop different components of the original model, so that we learn which components of the models are actually important for making the correct predictions.

MR:Yes, in further study we could try to improve the model with more recent environemental and surveillance data than in the original model.

References

1. Leighton PA, Koffi JK, Pelcat Y, Lindsay LR, Ogden NH. 2012. Predicting the speed of tick invasion: an empirical model of range expansion for the Lyme disease vector Ixodes scapularis in Canada. J Appl Ecol 49:457-464.

Reviewer #2: GENERAL REMARKS

The study validates a predictive model on tick distribution in Canada, by comparing collected field data with the model predictions. These types of model evaluations are useful to assess and improve on the predictive model. However, although I would consider this study to be worthwhile, there are many linguistic errors to be addressed before the publication of this manuscript. Please find more specific remarks below.

ABSTRACT

If the use of title words such as Background, Objective etc. was a conscious decision, these should be clearer. Perhaps adding “:” or a line break can make this distinction between different subheadings more readable.

Line 26: “The incidence of Lyme disease”, or “The recorded number of Lyme disease cases” is increasing in Québec…

Line 27: remove “,” after “2012”

Line 28: put “until 2100” after “establishment”

Line 44: I would add “Ixodes scapularis” to the list of key words

MR: done – changed in the text

INTRODUCTION

Line 49: Locally acquired cases (per year? In total?) please specify. You could say “increased from 144 annual cases in 2009, to 2636 in 2019”. Put “in Canada” at the beginning of the sentence.

Line 51: “is progressing”

Line 58: check commas and punctuation in general

Line 77: “Active acarological surveillance”

Line 95: all three stages

MR: done – changed in the text

METHODS

Study area: As this is a study assessing the viability of a model based on climatic data: Some general climatic information (mean yearly temperatures, precipitation), would be welcome. This is also true for the elevational range (as this is factored into the model), and overall vegetation composition (what type of forests).

MR:

Some general information on I. scapularis would be welcome, including the vectorial capacity of this species, as well as its ecology (questing behavior and host-parasite relations).

MR: changed in introduction l. 52-58

Line 104: “province”

Line 117: tick distribution “data”, comes from …

Line 123: which configuration? Lines, quadrats, one single 2000m line? Specify

Line 130: spacing after citation

Line 131: year of establishment

Line 132: years

MR: done – changed in the text

Line 139: “tick populations were”, or “a tick population was” supposed to be established

Line 138: remove first “of the model”, remove first comma

Line 146: “A tick population” or “tick populations”, there where and when…

Line 147: Predicted for it

Line 151: “if a tick population is predicted to be established in the area, then ticks are also expected to be found in the field data”

Line 169: for clarity, I suggest inserting a full equation in word in the form of:

Speed[km/year]=√(area*year*n)- √(area*year*n-1)

MR: done – changed in the text

Line 172: “exposed to ticks”, remove first comma,

Line 173: “predicted an established tick population”

Line 175: “across the population of Quebec”

Line 176: “…of established tick populations”, “was produced/generated”

MR: done – changed in the text

RESULTS

Line 184: add percentage

Line 199: “twice on average”

Line 209: “Concordance per collection”

MR: done – changed in the text

Table 1: Is there a reason why the three stages column data is in italic? If so, specify.

MR: yes, because the number of collections with three stages are included in tick presence (cf. footnote of table), it’s a sub-part of collection with tick presence

Table 2: If you use “Sd” in the table, the *text should be “Sd” as well, not “sd”.

Table 1 and 3: footnote: “are included”

Line 240: remove second “detection”

Line 242: “the three stages”

Line 245-246: If you include the statistical result, please also include the p-value

MR: done – changed in the text

DISCUSSION

I am missing some recommendations on if/how the findings of this study (field validation) could be used to enhance the model developed by Leighton (2012).

MR: added in the text Line 430

Line 298: “time-to-establishment”

Line 310: where and when “A tick population” or “tick populations” will establish

Line 315: Collect ticks

Line 320: Years

Line 321: remove comma. Risk of what? Specify Lyme disease risk

Line 322: “with the model predictions”

Line 338-339: Because of “limitations” in “logistical” capacity

Line 349: “influential factor”

Line 350: northern, sometimes “Quebec” is written “Québec”. Please be consistent.

MR: done – changed in the text

Line 351: host communities are different how?

MR: vertebrate host communities are different from southern Québec

Line 352: “A previous study”, or “previous studies”

Line 363: remove comma

MR: done – changed in the text

---

## [Decision Letter · Decision Letter 1]

25 Nov 2021

PONE-D-21-14260R1Current and future distribution of Ixodes scapularis ticks in Québec: field validation of a predictive modelPLOS ONE

Dear Dr. RIPOCHE,

Thank you for submitting your manuscript to PLOS ONE. After careful consideration, we feel that it has merit but does not fully meet PLOS ONE’s publication criteria as it currently stands. Therefore, we invite you to submit a revised version of the manuscript that addresses the points raised during the review process. I am weighting more heavily Reviewer 2's comments, who reviewed the original version and now recommends publication upon fixing a few minor details. However, please address Reviewer 3's concerns about statistical analyses as best as you can. Please ensure that your decision is justified on PLOS ONE’s publication criteria and not, for example, on novelty or perceived impact.

We look forward to receiving your revised manuscript.

Kind regards,

Catherine A. Brissette, Ph.D.

Academic Editor

PLOS ONE

Journal Requirements:

Reviewers' comments:

Reviewer's Responses to Questions

**Comments to the Author**

1. If the authors have adequately addressed your comments raised in a previous round of review and you feel that this manuscript is now acceptable for publication, you may indicate that here to bypass the “Comments to the Author” section, enter your conflict of interest statement in the “Confidential to Editor” section, and submit your "Accept" recommendation.

Reviewer #2: (No Response)

Reviewer #3: (No Response)

2. Is the manuscript technically sound, and do the data support the conclusions?

Reviewer #2: Yes

Reviewer #3: Partly

3. Has the statistical analysis been performed appropriately and rigorously? 

Reviewer #2: Yes

Reviewer #3: (No Response)

4. Have the authors made all data underlying the findings in their manuscript fully available?

Reviewer #2: Yes

Reviewer #3: (No Response)

5. Is the manuscript presented in an intelligible fashion and written in standard English?

Reviewer #2: Yes

Reviewer #3: Yes

6. Review Comments to the Author

Reviewer #2: GENERAL REMARKS

The proposed revised manuscript is a substantial improvement on the initial submission. Particularly, the discussion now thoroughly outlines the limitations of the findings and the methods, and how these may be improved in the future. Overall, I would consider this manuscript fit for publication, if a few minor changes are implemented.

Line 1: “TITLE”

INTRODUCTION

Line 51: “climatic” conditions

Line 52: …deciduous and mixed forests are “considered” suitable “habitats”…

Line 54: remove comma

Line 61: …drag sampling “method”.

Line 76: “northwards”

Line 97 and lower: If chosen to make et al. italic, please do so throughout the manuscript.

METHODS

Line 128-129: During the initial review, a request was made to specify the configuration of the 2000m2 area that was sampled using the flag dragging method. This was not done. It would be suitable to specify whether the sampling was done in simple line of 2000m, or if quadrats were used and if so, in which sampling structure.

Line 155-156: This question is repeating the same declarative statement of the sentence before. I suggest removing this question.

Line 184: Remove empty line to keep the spacing consistent with the other chapters

RESULTS:

Figure 2: Having black x and y labels would greatly enhance readability.

DISCUSSION

Of all the chapters, the discussion has improved the most compared to the initial submission.

Line 368: “limitations”

Reviewer #3: This study uses field data to validate a time-to-establishment model developed in 2012 by one of the authors to predict the Ixodes scapularis population establishment for each municipality in eastern Canada. This field data consisted of active tick surveillance data collected betweem 2010-2018, where two criteria were used to define tick establishment: i) either the detection of at least one tick or ii) the detection of the three questing stages of the tick.

Overall I really like the premise of this paper, dragging/flagging data are valuable but rarely get published, and I like that the authors have used a more creative/informative way of doing this by using it to validate a model (which in itself is also worthwhile). However there are some potentially significant issues with the use and interpretation of statistics, and some of the methods need more detail, for example how the human population increase was calculated and exactly when flagging/dragging (it is unclear which technique was used) was performed.

Introduction

The introduction does a good job of introducing the topic, highlights the importance of the paper and clearly sets out the aims of the study.

Line 48-54: All nice points! I would also argue that improved awareness of Lyme disease in Canada, and therefore improved diagnostics and reporting are also likely to have caused the number of human cases to increase.

Line 62: “In Québec, the number of 62 locally acquired human cases has increased from 2 cases in 2008 to 381 by 2019 (7)”

Human cases of what? Tick bite or Lyme disease? Clarify, and if Lyme disease I suggest moving this to line 58 after “a notifiable human disease in Québec since 2003.” for better information flow.

Lines 67-68: Which months are considered spring, summer and autumn in the study area (this is important when considering the months you sampled ticks).

Materials and Methods

Line 107: “The province of” should not be in bold

Line 122-132: There is some inconsistency in the description of tick surveillance, was it dragging with a drag cloth or flagging with a flag cloth (they can target different life stages of the tick). How frequently was flagging/dragging performed at each site? Line 126 you mention tick sampling in June-September, which life stages would you realistically be likely to encounter in each of these months at the study location? This information should be included as finding each of the life stages is an important underpinning of your analyses?

It is very hard for me to understand how the methods for sampling event concordance (line 146-152) and concordance over the study period (line 153-163) differ. After re-reading a few times I think I understood that concordance over the study period aggregated all tick data from 2010-2018 whereas sampling event concordance looked year by year? I think it would help the reader if this was made more explicit.

Line 165: Please clarify your use, justification, and interpretation of McNemar’s. this is not truly paired data and McNemar does not assess for concordance.

Line 177-179: “We cumulated the annual exposed human population from 2006 to 2100, and calculated the annual proportion of the Québec population exposed, based on the 2016 census data”. Cumulated and calculated how? Please add more detail.

Results

Line 202-204: This should be in the methods. Do I therefore understand correctly that sites were not sampled every year? In which case this also should be made more explicit in sampling event concordance (line 146-152) and concordance over the study period (line 153-163). It would be nice to have a table of when sites were sampled, with the raw tick data and what the model predicted (lower, mean and upper year).

Line 211-214: This really depends when sampling was performed (and another reason for wanting to see a table of exactly when sites were sampled and the raw tick data), as you mention in lines 67-68 the different life stages are active at very different times of year. So, if for example you happened to sample a bunch of municipalities in early spring when larvae and adults are unlikely to be active you are going to have a very different chance of finding all three life stages than if you sample in the summer, possibly biasing your results.

Line 306-311: It would be nice to add some further details here, for example, how many regions affected by Lyme disease? Is the coming years 1, 2, 5 or 10 years? By how much to neighbouring municipalities differ? Is there any pattern to this?

Figure 2: I propose changing the figure legend title to “proportion of the human population living in municipalities with established tick populations in Québec from 2008 to 2100” or similar, as any of the population could be “exposed” just by going one day for hike in a location with ticks.

Discussion

The general content of the discussion looks good and relevant, but I am hesitant at this stage to review it more thoroughly as I have concerns with the use and interpretation of statistics, and will be happy to read this in more detail once the authors have had a chance to respond to these concerns.

Line 323-325: “Concordance between observations and predictions was relatively low but statistically significant, with no major geographical inconsistencies” – I think you have interpreted the statistics incorrectly. The matched McNemar test does not test for concordance and I don’t see how these data are truly paired? Furthermore: “If the statistical significance level (i.e., p-value) is less than 0.05 (i.e., p < 0.05), you have a statistically significant result and the proportion of X before and after Y is statistically significantly different”, suggesting the opposite to concordance.

7. PLOS authors have the option to publish the peer review history of their article (what does this mean?). If published, this will include your full peer review and any attached files.

Reviewer #2: No

Reviewer #3: No

---

## [Author Response · Author response to Decision Letter 1]

7 Jan 2022

6. Review Comments to the Author

Reviewer #2: 

GENERAL REMARKS

The proposed revised manuscript is a substantial improvement on the initial submission. Particularly, the discussion now thoroughly outlines the limitations of the findings and the methods, and how these may be improved in the future. Overall, I would consider this manuscript fit for publication, if a few minor changes are implemented.

Line 1: “TITLE”

MR: done

INTRODUCTION

Line 51: “climatic” conditions

Line 52: …deciduous and mixed forests are “considered” suitable “habitats”…

Line 54: remove comma

Line 61: …drag sampling “method”.

Line 76: “northwards”

MR: done

Line 97 and lower: If chosen to make et al. italic, please do so throughout the manuscript.

MR: done

METHODS

Line 128-129: During the initial review, a request was made to specify the configuration of the 2000m2 area that was sampled using the flag dragging method. This was not done. It would be suitable to specify whether the sampling was done in simple line of 2000m, or if quadrats were used and if so, in which sampling structure.

MR: done. I added explanation in text, line 130-131

Line 155-156: This question is repeating the same declarative statement of the sentence before. I suggest removing this question.

MR: I prefer to keep the question to clarify for the lector, explaining the analysis in another way

Line 184: Remove empty line to keep the spacing consistent with the other chapters

MR: I standardized between sections

RESULTS:

Figure 2: Having black x and y labels would greatly enhance readability.

MR: done

DISCUSSION

Of all the chapters, the discussion has improved the most compared to the initial submission.

MR: Thanks

Line 368: “limitations”

MR: done

Reviewer #3: 

This study uses field data to validate a time-to-establishment model developed in 2012 by one of the authors to predict the Ixodes scapularis population establishment for each municipality in eastern Canada. This field data consisted of active tick surveillance data collected betweem 2010-2018, where two criteria were used to define tick establishment: i) either the detection of at least one tick or ii) the detection of the three questing stages of the tick.

Overall I really like the premise of this paper, dragging/flagging data are valuable but rarely get published, and I like that the authors have used a more creative/informative way of doing this by using it to validate a model (which in itself is also worthwhile). 

MR: Thanks

However there are some potentially significant issues with the use and interpretation of statistics, and some of the methods need more detail, for example how the human population increase was calculated and exactly when flagging/dragging (it is unclear which technique was used) was performed.

MR: I tried to clarify that in the text

Introduction

The introduction does a good job of introducing the topic, highlights the importance of the paper and clearly sets out the aims of the study.

MR: Thanks

Line 48-54: All nice points! I would also argue that improved awareness of Lyme disease in Canada, and therefore improved diagnostics and reporting are also likely to have caused the number of human cases to increase.

MR: I agree with you

Line 62: “In Québec, the number of 62 locally acquired human cases has increased from 2 cases in 2008 to 381 by 2019 (7)”

Human cases of what? Tick bite or Lyme disease? Clarify, and if Lyme disease I suggest moving this to line 58 after “a notifiable human disease in Québec since 2003.” for better information flow.

MR: done. 

Lines 67-68: Which months are considered spring, summer and autumn in the study area (this is important when considering the months you sampled ticks).

MR: done.

Materials and Methods

Line 107: “The province of” should not be in bold

MR: done.

Line 122-132: There is some inconsistency in the description of tick surveillance, was it dragging with a drag cloth or flagging with a flag cloth (they can target different life stages of the tick). 

MR: I added details about dragging methods

How frequently was flagging/dragging performed at each site? 

MR: I added details and there is also information in results, line 209-210

Line 126 you mention tick sampling in June-September, which life stages would you realistically be likely to encounter in each of these months at the study location? This information should be included as finding each of the life stages is an important underpinning of your analyses?

MR: I added details

It is very hard for me to understand how the methods for sampling event concordance (line 146-152) and concordance over the study period (line 153-163) differ. After re-reading a few times I think I understood that concordance over the study period aggregated all tick data from 2010-2018 whereas sampling event concordance looked year by year? I think it would help the reader if this was made more explicit.

MR: yes, that is the difference. I added some details in the text.

Line 165: Please clarify your use, justification, and interpretation of McNemar’s. this is not truly paired data and McNemar does not assess for concordance.

MR: done. Line 172: “A significant Chi-square test, with p<0.05, suggests that the distribution of sites with presence of ticks is not at random between areas with expected or unexpected established tick population according to the model. We used the matched McNemar test for the analysis of all the sampling because of some repeated samples at the same site”

Line 177-179: “We cumulated the annual exposed human population from 2006 to 2100, and calculated the annual proportion of the Québec population exposed, based on the 2016 census data”. Cumulated and calculated how? Please add more detail.

MR: done

Results

Line 202-204: This should be in the methods. Do I therefore understand correctly that sites were not sampled every year? In which case this also should be made more explicit in sampling event concordance (line 146-152) and concordance over the study period (line 153-163). It would be nice to have a table of when sites were sampled, with the raw tick data and what the model predicted (lower, mean and upper year).

MR: I added some details in methods. I let description of data in result. Data are available is a request to specific person.

Line 211-214: This really depends when sampling was performed (and another reason for wanting to see a table of exactly when sites were sampled and the raw tick data), as you mention in lines 67-68 the different life stages are active at very different times of year. So, if for example you happened to sample a bunch of municipalities in early spring when larvae and adults are unlikely to be active you are going to have a very different chance of finding all three life stages than if you sample in the summer, possibly biasing your results.

MR: Sites are sampling generally once, between may and September. In endemic area, we are able to catch nymph, adults and larvae in spring. In discussion, this is the main limitation of active surveillance data (line 369), and that is why we assess the concordance with presence of tick and not only presence of confirmed established tick population.

Line 306-311: It would be nice to add some further details here, for example, how many regions affected by Lyme disease? Is the coming years 1, 2, 5 or 10 years? By how much to neighbouring municipalities differ? Is there any pattern to this?

MR: I added some details in the text (Line 310). We didn’t analyse the pattern of expansion, because it was outside the scope of our study.

Figure 2: I propose changing the figure legend title to “proportion of the human population living in municipalities with established tick populations in Québec from 2008 to 2100” or similar, as any of the population could be “exposed” just by going one day for hike in a location with ticks.

MR: done

Discussion

The general content of the discussion looks good and relevant, but I am hesitant at this stage to review it more thoroughly as I have concerns with the use and interpretation of statistics, and will be happy to read this in more detail once the authors have had a chance to respond to these concerns.

Line 323-325: “Concordance between observations and predictions was relatively low but statistically significant, with no major geographical inconsistencies” – I think you have interpreted the statistics incorrectly. The matched McNemar test does not test for concordance and I don’t see how these data are truly paired? Furthermore: “If the statistical significance level (i.e., p-value) is less than 0.05 (i.e., p < 0.05), you have a statistically significant result and the proportion of X before and after Y is statistically significantly different”, suggesting the opposite to concordance.

MR: see explanation in methods. We assessed concordance of presence of tick population between prediction and observation. We tested concordance in predicted area vs unpredicted area, for random distribution (chi-square test) ans level of concordance (kappa).

7. PLOS authors have the option to publish the peer review history of their article (what does this mean?). If published, this will include your full peer review and any attached files.

Do you want your identity to be public for this peer review? For information about this choice, including consent withdrawal, please see our Privacy Policy.

Reviewer #2: No

Reviewer #3: No

---

## [Editor Report · Decision Letter 2]

18 Jan 2022

Current and future distribution of Ixodes scapularis ticks in Québec: field validation of a predictive model

PONE-D-21-14260R2

Dear Dr. RIPOCHE,

We’re pleased to inform you that your manuscript has been judged scientifically suitable for publication and will be formally accepted for publication once it meets all outstanding technical requirements.

Kind regards,

Catherine A. Brissette, Ph.D.

Academic Editor

PLOS ONE
---

## [Editor Report · Acceptance letter]

25 Jan 2022

PONE-D-21-14260R2 

Current and future distribution of *Ixodes scapularis* ticks in Québec: field validation of a predictive model 

Dear Dr. Ripoche:

I'm pleased to inform you that your manuscript has been deemed suitable for publication in PLOS ONE. Congratulations! Your manuscript is now with our production department. 

Kind regards, 

on behalf of

Dr. Catherine A. Brissette 

Academic Editor

PLOS ONE